# Regressive Virtual Metric Learning

**Michaël Perrot, and Amaury Habrard**
Université de Lyon, Université Jean Monnet de Saint-Etienne,
Laboratoire Hubert Curien, CNRS, UMR5516, F-42000, Saint-Etienne, France.
`{michael.perrot,amaury.habrard}@univ-st-etienne.fr`

## Abstract

We are interested in supervised metric learning of Mahalanobis like distances. Existing approaches mainly focus on learning a new distance using similarity and dissimilarity constraints between examples. In this paper, instead of bringing closer examples of the same class and pushing far away examples of different classes we propose to move the examples with respect to virtual points. Hence, each example is brought closer to a a priori defined virtual point reducing the number of constraints to satisfy. We show that our approach admits a closed form solution which can be kernelized. We provide a theoretical analysis showing the consistency of the approach and establishing some links with other classical metric learning methods. Furthermore we propose an efficient solution to the difficult problem of selecting virtual points based in part on recent works in optimal transport. Lastly, we evaluate our approach on several state of the art datasets.

## 1   Introduction

The goal of a metric learning algorithm is to capture the idiosyncrasies in the data mainly by defining a new space of representation where some semantic constraints between examples are fulfilled. In the previous years the main focus of metric learning algorithms has been to learn Mahalanobis like distances of the form $d_{\mathbf{M}}(\mathbf{x}, \mathbf{x}') = \sqrt{(\mathbf{x} - \mathbf{x}')^T \mathbf{M}(\mathbf{x} - \mathbf{x}')}$ where $\mathbf{M}$ is a positive semi-definite matrix (PSD) defining a set of parameters[1]. Using a Cholesky decomposition $\mathbf{M} = \mathbf{L}\mathbf{L}^T$, one can see that this is equivalent to learn a linear transformation from the input space.

Most of the existing approaches in metric learning use constraints of type must-link and cannot-link between learning examples [1, 2]. For example, in a supervised classification task, the goal is to bring closer examples of the same class and to push far away examples of different classes. The idea is that the learned metric should affect a high value to dissimilar examples and a low value to similar examples. Then, this new distance can be used in a classification algorithm like a nearest neighbor classifier. Note that in this case the set of constraints is quadratic in the number of examples which can be prohibitive when the number of examples increases. One heuristic is then to select only a subset of the constraints but selecting such a subset is not trivial. In this paper, we propose to consider a new kind of constraints where each example is associated with an a priori defined virtual point. It allows us to consider the metric learning problem as a simple regression where we try to minimize the differences between learning examples and virtual points. Fig. 1 illustrates the differences between our approach and a classical metric learning approach. It can be noticed that our algorithm only uses a linear number of constraints. However defining these constraints by hand can be tedious and difficult. To overcome this problem, we present two approaches to automatically define them. The first one is based on some recent advances in the field of Optimal Transport while the second one uses a class-based representation space.

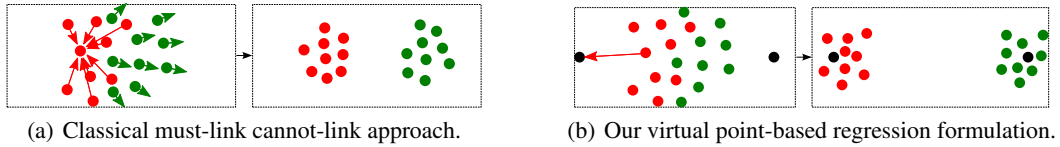

(a) Classical must-link cannot-link approach.          (b) Our virtual point-based regression formulation.

Figure 1: Arrows denote the constraints used by each approach for one particular example in a binary classification task. The classical metric learning approach in Fig. 1(a) uses $\mathcal{O}(n^2)$ constraints bringing closer examples of the same class and pushing far away examples of different classes. On the contrary, our approach presented in Fig. 1(b) moves the examples to the neighborhood of their corresponding virtual point, in black, using only $\mathcal{O}(n)$ constraints. ( Best viewed in color )

Moreover, thanks to its regression-based formulation, our approach can be easily kernelized allowing us to deal efficiently with non linear transformations which is a nice advantage in comparison to some metric learning methods. We also provide a theoretical analysis showing the consistency of our approach and establishing some relationships with a classical metric learning formulation.

This paper is organized as follows. In Section 2 we identify several related works. Then in Section 3 we present our approach, provide some theoretical results and give two solutions to generate the virtual points. Section 4 is dedicated to an empirical evaluation of our method on several widely used datasets. Finally, we conclude in Section 5.

## 2   Related work

For up-to-date surveys on metric learning see [3] and [4]. In this section we focus on algorithms which are more closely related to our approach. First of all, one of the most famous approach in metric learning is LMNN [5] where the authors propose to learn a PSD matrix to improve the k-nearest-neighbours algorithm. In their work, instead of considering pairs of examples, they use triplets $(\mathbf{x}_i, \mathbf{x}_j, \mathbf{x}_k)$ where $\mathbf{x}_j$ and $\mathbf{x}_k$ are in the neighborhood of $\mathbf{x}_i$ and such that $\mathbf{x}_i$ and $\mathbf{x}_j$ are of the same class and $\mathbf{x}_k$ is of a different class. The idea is then to bring closer $\mathbf{x}_i$ and $\mathbf{x}_j$ while pushing $\mathbf{x}_k$ far away. Hence, if the number of constraints seems to be cubic, the authors propose to only consider triplets of examples which are already close to each other. In contrast, the idea presented in [6] is to collapse all the examples of the same class in a single point and to push infinitely far away examples of different classes. The authors define a measure to estimate the probability of having an example $\mathbf{x}_j$ given an example $\mathbf{x}_i$ with respect to a learned PSD matrix $\mathbf{M}$. Then, they minimize, *w.r.t.* $\mathbf{M}$, the KL divergence between this measure and the best case where the probability is 1 if the two examples are of the same class and 0 otherwise. It can be seen as collapsing all the examples of the same class on an implicit virtual point. In this paper we use several explicit virtual points and we collapse the examples on these points with respect to their classes and their distances to them.

A recurring issue in Mahalanobis like metric learning is to fulfill the PSD constraint on the learned metric. Indeed, projecting a matrix on the PSD cone is not trivial and generally requires a costly eigenvalues decomposition. To address this problem, in ITML [1] the authors propose to use a LogDet divergence as the regularization term. The idea is to learn a matrix which is close to an a priori defined PSD matrix. The authors then show that if the divergence is finite, then the learned matrix is guaranteed to be PSD. Another approach, as proposed in [2], is to learn a matrix $\mathbf{L}$ such that $\mathbf{M} = \mathbf{L}\mathbf{L}^T$, *i.e.* instead of learning the metric the authors propose to learn the projection. The main drawback is the fact that most of the time the resulting optimization problem is not convex [3, 4, 7] and is thus harder to optimize. In this paper, we are also interested in learning $\mathbf{L}$ directly. However, because we are using constraints between examples and virtual points, we obtain a convex problem with a closed form solution allowing us to learn the metric in an efficient way.

The problem of learning a metric such that the induced space is not linearly dependent of the input space has been addressed in several works before. First, it is possible to directly learn an intrinsically non linear metric as in $\chi^2$-LMNN [8] where the authors propose to learn a $\chi^2$ distance rather than a Mahalanobis distance. This distance is particularly relevant for histograms comparisons. Note that this kind of approaches is close to the kernel learning problem which is beyond the scope of this work. Second, another solution used by local metric learning methods is to split the input space

in several regions and to learn a metric in each region to introduce some non linearity, as in MM-LMNN [7]. Similarly, in GB-LMNN [8] the authors propose to locally refine the metric learned by LMNN by successively splitting the input space. A third kind of approach tries to project the learning examples in a new space which is non linearly dependent of the input space. It can be done in two ways, either by projecting a priori the learning examples in a new space with a KPCA [9] or by rewriting the optimization problem in a kernelized form [1]. The first approach allows one to include non linearity in most of the metric learning algorithms but imposes to select the interesting features beforehand. The second method can be difficult to use as rewriting the optimization problem is most of the times non trivial [4]. Indeed, if one wants to use the kernel trick it implies that the access to the learning examples should only be done through dot products which is difficult when working with pairs of examples as it is the case in metric learning. In this paper we show that using virtual points chosen in a given target space allows us to kernelize our approach easily and thus to work in a very high dimensional space without using an explicit projection thanks to the kernel trick.

Our method is based on a regression and can thus be linked, in its kernelized form, to several approaches in kernelized regression for structured output [10, 11, 12]. The idea behind these approaches is to minimize the difference between input examples and output examples using kernels, *i.e.* working in a high dimensional space. In our case, the learning examples can be seen as input examples and the virtual points as output examples. However, we only project the learning examples in a high dimensional space, the virtual points already belong to the output space. Hence, we do not have the pre-image problem [12]. Furthermore, our goal is not to predict a virtual point but to learn a metric between examples and thus, after the learning step, the virtual points are discarded.

## 3 Contributions

The main idea behind our algorithm is to bring closer the learning examples to a set of virtual points. We present this idea in three subsections. First we assume that we have access to a set of $n$ learning pairs $(\mathbf{x},\mathbf{v})$ where $\mathbf{x}$ is a learning example and $\mathbf{v}$ is a virtual point associated to $\mathbf{x}$ and we present both the linear and kernelized formulations of our approach called RVML. It boils down to solve a regression in closed form, the main originality being the introduction of virtual points. In the second subsection, we show that it is possible to theoretically link our approach to a classical metric learning one based on [13]. In the last subsection, we propose two automatic methods to generate the virtual points and to associate them with the learning examples.

### 3.1 Regressive Virtual Metric Learning (RVML)

Given a probability distribution $\mathcal{D}$ defined over $\mathcal{X} \times \mathcal{Y}$ where $\mathcal{X} \subseteq \mathbb{R}^d$ and $\mathcal{Y}$ is a finite label set, let $S = \{(\mathbf{x}_i, y_i)\}_{i=1}^n$ be a set of examples drawn i.i.d. from $\mathcal{D}$. Let $f_{\mathbf{v}} : \mathcal{X} \times \mathcal{Y} \to \mathcal{V}$ where $\mathcal{V} \subseteq \mathbb{R}^{d'}$ be the function which associates each example to a virtual point. We consider the learning set $S_{\mathbf{v}} = \{(\mathbf{x}_i, \mathbf{v}_i)\}_{i=1}^n$ where $\mathbf{v}_i = f_{\mathbf{v}}(\mathbf{x}_i, y_i)$. For the sake of simplicity denote by $\mathbf{X} = (\mathbf{x}_1, \dots, \mathbf{x}_n)^T$ and $\mathbf{V} = (\mathbf{v}_1, \dots, \mathbf{v}_n)^T$ the matrices containing respectively one example and the associated virtual point on each line. In this section, we consider that the function $f_{\mathbf{v}}$ is known. We come back to its definition in Section 3.3. Let $\| \cdot \|_{\mathcal{F}}$ be the Frobenius norm and $\| \cdot \|_2$ be the $l_2$ vector norm. Our goal is to learn a matrix $\mathbf{L}$ such that $\mathbf{M} = \mathbf{L}\mathbf{L}^T$ and for this purpose we consider the following optimisation problem:

$$\min_{\mathbf{L}} f(\mathbf{L}, \mathbf{X}, \mathbf{V}) = \min_{\mathbf{L}} \frac{1}{n} \|\mathbf{X}\mathbf{L} - \mathbf{V}\|_{\mathcal{F}}^2 + \lambda \|\mathbf{L}\|_{\mathcal{F}}^2. \tag{1}$$

The idea is to learn a new space of representation where each example is close to its associated virtual point. Note that $\mathbf{L}$ is a $d \times d'$ matrix and if $d' < d$ we also perform dimensionality reduction.

**Theorem 1.** *The optimal solution of Problem 1 can be found in closed form. Furthermore, we can derive two equivalent solutions:*

$$\mathbf{L} = \left(\mathbf{X}^T\mathbf{X} + \lambda n \mathbf{I}\right)^{-1} \mathbf{X}^T \mathbf{V} \tag{2}$$

$$\mathbf{L} = \mathbf{X}^T \left(\mathbf{X}\mathbf{X}^T + \lambda n \mathbf{I}\right)^{-1} \mathbf{V}. \tag{3}$$

*Proof.* The proof of this theorem can be found in the supplementary material. □

From Eq. 2 we deduce the matrix $\mathbf{M}$:

$$\mathbf{M} = \mathbf{L}\mathbf{L}^T = \left(\mathbf{X}^T\mathbf{X} + \lambda n\mathbf{I}\right)^{-1}\mathbf{X}^T\mathbf{V}\mathbf{V}^T\mathbf{X}\left(\mathbf{X}^T\mathbf{X} + \lambda n\mathbf{I}\right)^{-1}. \tag{4}$$

Note that $\mathbf{M}$ is PSD by construction: $\mathbf{x}^T\mathbf{M}\mathbf{x} = \mathbf{x}^T\mathbf{L}\mathbf{L}^T\mathbf{x} = \|\mathbf{L}^T\mathbf{x}\|_2^2 \geq 0$.

So far, we have focused on the linear setting. We now present a kernelized version, showing that it is possible to learn a metric in a very high dimensional space without an explicit projection.

Let $\phi(\mathbf{x})$ be a projection function and $K(\mathbf{x}, \mathbf{x}') = \phi(\mathbf{x})^T\phi(\mathbf{x}')$ be its associated kernel. For the sake of readability, let $K_{\mathbf{X}} = \phi(\mathbf{X})\phi(\mathbf{X})^T$ where $\phi(\mathbf{X}) = (\phi(\mathbf{x}_1), \ldots, \phi(\mathbf{x}_n))^T$. Given the solution matrix $\mathbf{L}$ presented in Eq. 3, we have $\mathbf{M} = \mathbf{X}^T\left(\mathbf{X}\mathbf{X}^T + \lambda n\mathbf{I}\right)^{-1}\mathbf{V}\mathbf{V}^T\left(\mathbf{X}\mathbf{X}^T + \lambda n\mathbf{I}\right)^{-1}\mathbf{X}$. Then, $\mathbf{M}_K$ the kernelized version of the matrix $\mathbf{M}$ is defined such that:

$$\mathbf{M}_K = \phi(\mathbf{X})^T\left(K_{\mathbf{X}} + \lambda n\mathbf{I}\right)^{-1}\mathbf{V}\mathbf{V}^T\left(K_{\mathbf{X}} + \lambda n\mathbf{I}\right)^{-1}\phi(\mathbf{X}).$$

The squared Mahalanobis distance can be written as $d_{\mathbf{M}}^2(\mathbf{x}, \mathbf{x}') = \mathbf{x}^T\mathbf{M}\mathbf{x} + \mathbf{x}'^T\mathbf{M}\mathbf{x}' - 2\mathbf{x}^T\mathbf{M}\mathbf{x}'$. Thus we can obtain $d_{\mathbf{M}_K}^2(\phi(\mathbf{x}), \phi(\mathbf{x}')) = \phi(\mathbf{x})^T\mathbf{M}_K\phi(\mathbf{x}) + \phi(\mathbf{x}')^T\mathbf{M}_K\phi(\mathbf{x}') - 2\phi(\mathbf{x})^T\mathbf{M}_K\phi(\mathbf{x}')$ the kernelized version by considering that:

$$\phi(\mathbf{x})^T\mathbf{M}_K\phi(\mathbf{x}) = \phi(\mathbf{x})^T\phi(\mathbf{X})^T\left(K_{\mathbf{X}} + \lambda n\mathbf{I}\right)^{-1}\mathbf{V}\mathbf{V}^T\left(K_{\mathbf{X}} + \lambda n\mathbf{I}\right)^{-1}\phi(\mathbf{X})\phi(\mathbf{x})$$
$$= K_{\mathbf{X}}(\mathbf{x})^T\left(K_{\mathbf{X}} + \lambda n\mathbf{I}\right)^{-1}\mathbf{V}\mathbf{V}^T\left(K_{\mathbf{X}} + \lambda n\mathbf{I}\right)^{-1}K_{\mathbf{X}}(\mathbf{x})$$

where $K_{\mathbf{X}}(\mathbf{x}) = (K(\mathbf{x}, \mathbf{x}_1), \ldots, K(\mathbf{x}, \mathbf{x}_n))^T$ is the similarity vector to the examples *w.r.t.* $K$.

Note that it is also possible to obtain a kernelized version of $\mathbf{L}$: $\mathbf{L}_K = \phi(\mathbf{X})^T\left(K_{\mathbf{X}} + \lambda n\mathbf{I}\right)^{-1}\mathbf{V}$.

This result is close to a previous one already derived in [11] in a structured output setting. The main difference is the fact that we do not use a kernel on the output (the virtual points here). Hence, it is possible to compute the projection of an example $\mathbf{x}$ of dimension $d$ in a new space of dimension $d'$:

$$\phi(\mathbf{x})\mathbf{L}_K = \phi(\mathbf{x})^T\phi(\mathbf{X})^T\left(K_{\mathbf{X}} + \lambda n\mathbf{I}\right)^{-1}\mathbf{V} = K_{\mathbf{X}}(\mathbf{x})^T\left(K_{\mathbf{X}} + \lambda n\mathbf{I}\right)^{-1}\mathbf{V}.$$

Recall that in this work we are interested in learning a distance between examples and not in the prediction of the virtual points which only serve as a way to bring closer similar examples and push far away dissimilar examples.

From a complexity standpoint, we can see that, assuming the kernel function as easy to calculate, the main bottleneck when computing the solution in closed form is the inversion of a $n \times n$ matrix.

## 3.2 Theoretical Analysis

In this section, we propose to theoretically show the interest of our approach by proving (i) that it is consistent and (ii) that it is possible to link it to a more classical metric learning formulation.

### 3.2.1 Consistency

Let $l(\mathbf{L}, (\mathbf{x}, \mathbf{v})) = \|\mathbf{x}^T\mathbf{L} - \mathbf{v}^T\|_2^2$ be our loss and let $\mathcal{D}_{\mathbf{v}}$ be the probability distribution over $\mathcal{X} \times \mathcal{V}$ such that $p_{\mathcal{D}_{\mathbf{v}}}(\mathbf{x}, \mathbf{v}) = p_{\mathcal{D}}(\mathbf{x}, y|\mathbf{v} = f_{\mathbf{v}}(\mathbf{x}, y))$. Showing the consistency boils down to bound with high probability the true risk, denoted by $R(\mathbf{L})$, by the empirical risk, denoted by $\hat{R}(\mathbf{L})$ such that:

$$R(\mathbf{L}) = \mathbb{E}_{(\mathbf{x}, \mathbf{v}) \sim \mathcal{D}_{\mathbf{v}}} l(\mathbf{L}, (\mathbf{x}, \mathbf{v})) \text{ and } \hat{R}(\mathbf{L}) = \frac{1}{n}\sum_{(\mathbf{x}, \mathbf{v}) \in S_{\mathbf{v}}} l(\mathbf{L}, (\mathbf{x}, \mathbf{v})) = \frac{1}{n}\|\mathbf{X}\mathbf{L} - \mathbf{V}\|_{\mathcal{F}}^2.$$

The empirical risk corresponds to the error of the learned matrix $\mathbf{L}$ on the learning set $S_{\mathbf{v}}$. The true risk is the error of $\mathbf{L}$ on the unknown distribution $\mathcal{D}_{\mathbf{v}}$. The consistency property ensures that with a sufficient number of examples a low empirical risk implies a low true risk with high probability. To show that our approach is consistent, we use the uniform stability framework [14].

**Theorem 2.** *Let $\|\mathbf{v}\|_2 \leq C_{\mathbf{v}}$ for any $\mathbf{v} \in \mathcal{V}$ and $\|\mathbf{x}\|_2 \leq C_{\mathbf{x}}$ for any $\mathbf{x} \in \mathcal{X}$. With probability $1 - \delta$, for any matrix $\mathbf{L}$ optimal solution of Problem 1, we have:*

$$R(\mathbf{L}) \leq \hat{R}(\mathbf{L}) + \frac{8C_{\mathbf{v}}^2 C_{\mathbf{x}}^2}{\lambda n}\left(1 + \frac{C_{\mathbf{x}}}{\sqrt{\lambda}}\right)^2 + \left(\left(\frac{16C_{\mathbf{x}}^2}{\lambda} + 1\right)C_{\mathbf{v}}^2\left(1 + \frac{C_{\mathbf{x}}}{\sqrt{\lambda}}\right)^2\right)\sqrt{\frac{\ln\frac{1}{\delta}}{2n}}.$$

*Proof.* The proof of this theorem can be found in the supplementary material. ☐

We obtain a rate of convergence in $\mathcal{O}\left(\frac{1}{\sqrt{n}}\right)$ which is standard with this kind of bounds.

### 3.2.2 Link with a Classical Metric Learning Formulation

In this section we show that it is possible to bound the true risk of a classical metric learning approach with the empirical risk of our formulation. Most of the classical metric learning approaches make use of a notion of margin between similar and dissimilar examples. Hence, similar examples have to be close to each other, *i.e.* at a distance smaller than a margin $\gamma_1$, and dissimilar examples have to be far from each other, *i.e.* at a distance greater than a margin $\gamma_{-1}$. Let $(\mathbf{x}_i, y_i)$ and $(\mathbf{x}_j, y_j)$ be two examples from $\mathcal{X} \times \mathcal{Y}$, using this notion of margin, we consider the following loss [13]:

$$l(\mathbf{L}, (\mathbf{x}_i, y_i), (\mathbf{x}_j, y_j)) = \left[ y_{ij}(d^2(\mathbf{L}^T \mathbf{x}_i, \mathbf{L}^T \mathbf{x}_j) - \gamma_{y_{ij}}) \right]_+ \tag{5}$$

where $y_{ij} = 1$ if $y_i = y_j$ and $-1$ otherwise, $[z]_+ = \max(0, z)$ is the hinge loss and $\gamma_{y_{ij}}$ is the desired margin between examples. As introduced before, we consider that $\gamma_{y_{ij}}$ takes a big value when the examples are dissimilar, *i.e.* when $y_{ij} = -1$, and a small value when the examples are similar, *i.e.* when $y_{ij} = 1$. In the following we show that, relating the notion of margin to the distances between virtual points, it is possible to bound the true risk associated with this loss by the empirical risk of our approach with respect to a constant.

**Theorem 3.** *Let $\mathcal{D}$ be a distribution over $\mathcal{X} \times \mathcal{Y}$. Let $\mathcal{V} \subset \mathbb{R}^{d'}$ be a finite set of virtual points and $f_{\mathbf{v}}$ is defined as $f_{\mathbf{v}}(\mathbf{x}_i, y_i) = \mathbf{v}_i$, $\mathbf{v}_i \in \mathcal{V}$. Let $\|\mathbf{v}\|_2 \leq C_{\mathbf{v}}$ for any $\mathbf{v} \in \mathcal{V}$ and $\|\mathbf{x}\|_2 \leq C_{\mathbf{x}}$ for any $\mathbf{x} \in \mathcal{X}$. Let $\gamma_1 = 2\max_{\mathbf{x}_k, \mathbf{x}_l, y_{kl}=1} d^2(\mathbf{v}_k, \mathbf{v}_l)$ and $\gamma_{-1} = \frac{1}{2}\min_{\mathbf{x}_k, \mathbf{x}_l, y_{kl}=-1} d^2(\mathbf{v}_k, \mathbf{v}_l)$, we have:*

$$\mathbb{E}_{(\mathbf{x}_i, y_i) \sim \mathcal{D}, (\mathbf{x}_j, y_j) \sim \mathcal{D}} \left[ y_{ij}(d^2(\mathbf{L}^T \mathbf{x}_i, \mathbf{L}^T \mathbf{x}_j) - \gamma_{y_{ij}}) \right]_+$$

$$\leq 8 \left( \hat{R}(\mathbf{L}) + \frac{8C_{\mathbf{v}}^2 C_{\mathbf{x}}^2}{\lambda n} \left( 1 + \frac{C_{\mathbf{x}}}{\sqrt{\lambda}} \right)^2 + \left( \left( \frac{16C_{\mathbf{x}}^2}{\lambda} + 1 \right) C_{\mathbf{v}}^2 \left( 1 + \frac{C_{\mathbf{x}}}{\sqrt{\lambda}} \right)^2 \right) \sqrt{\frac{\ln \frac{1}{\delta}}{2n}} \right).$$

*Proof.* The proof of this theorem can be found in the supplementary material. ☐

In Theorem 3, we can notice that the margins are related to the distances between virtual points and correspond to the ideal margins, *i.e.* the margins that we would like to achieve after the learning step. Aside this remark, we can define $\hat{\gamma}_1$ and $\hat{\gamma}_{-1}$ the observed margins obtained after the learning step: All the similar examples are in a sphere centered in their corresponding virtual point and of diameter $\hat{\gamma}_1 = 2\max_{(\mathbf{x}, \mathbf{v})} \|\mathbf{x}^T \mathbf{L} - \mathbf{v}^T\|_2$. Similarly, the distance between hyperspheres of dissimilar examples is $\hat{\gamma}_{-1} = \min_{\mathbf{v}, \mathbf{v}', \mathbf{v} \neq \mathbf{v}'} \|\mathbf{v} - \mathbf{v}'\|_2 - \hat{\gamma}_1$. As a consequence, even if we do not use cannot-link constraints our algorithm is able to push reasonably far away dissimilar examples.

In the next subsection we present two different methods to select the virtual points.

### 3.3 Virtual Points Selection

Previously, we assumed to have access to the function $f_{\mathbf{v}} : \mathcal{X} \times \mathcal{Y} \to \mathcal{V}$. In this subsection, we present two methods for generating automatically the set of virtual points and the mapping $f_{\mathbf{v}}$.

#### 3.3.1 Using Optimal Transport on the Learning Set

In this first approach, we propose to generate the virtual points by using a recent variation of the Optimal Transport (OT) problem [15] allowing one to transport some examples to new points corresponding to a linear combination of a set of known instances. These new points will actually correspond to our virtual points. Our approach works as follows. We begin by extracting a set of landmarks $S'$ from the training set $S$. For this purpose, we use an adaptation of the landmark selection method proposed in [16] allowing us to take into account some diversity among the landmarks. To avoid to fix the number of landmarks in advance, we have just replaced it by a simple heuristic saying that the number of landmarks must be greater than the number of classes and that the maximum distance between an example and a landmark must be lower than the mean of all pairwise

**Algorithm 1:** Selecting $S'$ from a set of examples $S$.

---

**input** : $S = \{(\mathbf{x}_i, y_i)\}_{i=1}^{n}$ a set of examples; $\mathcal{Y}$ the label set.

**output**: $S'$ a subset of $S$

**begin**

    $\mu$ = mean of distances between all the examples of $S$

    $\mathbf{x}_{\max} = \arg\max\limits_{\mathbf{x} \in S} \|\mathbf{x} - \mathbf{0}\|_2$

    $S' = \{\mathbf{x}_{\max}\}; S = S \setminus S'$

    $\varepsilon = \max_{\mathbf{x} \in S} \min_{\mathbf{x}' \in S'} \|\mathbf{x} - \mathbf{x}'\|_2$

    **while** $|S'| < |\mathcal{Y}|$ **or** $\varepsilon > \mu$ **do**

        $\mathbf{x}_{\max} = \arg\max\limits_{\mathbf{x} \in S} \sum\limits_{\mathbf{x}' \in S'} \|\mathbf{x} - \mathbf{x}'\|_2$

        $S' = S' \cup \{\mathbf{x}_{\max}\}; S = S \setminus S'$

        $\varepsilon = \max_{\mathbf{x} \in S} \min_{\mathbf{x}' \in S'} \|\mathbf{x} - \mathbf{x}'\|_2$

---

distances from the training set -allowing us to have a fully automatic procedure. It is summarized in Algorithm 1.

Then we compute an optimal transport from the training set $S$ to the landmark set $S'$. For this purpose, we create a real matrix $\mathbf{C}$ of size $|S| \times |S'|$ giving the cost to transport one training instance to a landmark such that $\mathbf{C}(i,j) = \|\mathbf{x}_i - \mathbf{x}'_j\|_2$ with $\mathbf{x}_i \in S$ and $\mathbf{x}'_j \in S'$. The optimal transport is found by learning a matrix $\gamma \in \mathbb{R}^{|S| \times |S'|}$ able to minimize the cost of moving training examples to the landmark points. Let $\mathbf{S}'$ be the matrix of landmark points (one per line), the transport *w.r.t.* $\gamma$ of any training instance $(\mathbf{x}_i, y_i)$ gives a new virtual point such that $f_\mathbf{v}(\mathbf{x}_i, y_i) = \gamma(i)\mathbf{S}'$, $\gamma(i)$ designing the $i^{\text{th}}$ line of $\gamma$. Note that this new virtual point is a linear combination of the landmark instances to which the example is transported. The set of virtual points is then defined by $\mathbf{V} = \gamma\mathbf{S}'$. The virtual points are thus not defined a priori but are automatically learned by solving a problem of optimal transport. Note that this transportation mode is potentially non linear since there is no guarantee that there exists a matrix $\mathbf{T}$ such that $\mathbf{V} = \mathbf{XT}$. Our metric learning approach can, in this case, be seen as a an approximation of the result given by the optimal transport.

To learn $\gamma$, we use the following optimization problem proposed in [17]:

$$\arg\min_{\gamma} \langle \gamma, \mathbf{C} \rangle_{\mathcal{F}} - \frac{1}{\lambda} h(\gamma) + \eta \sum_j \sum_c \|\gamma(y_i = c, j)\|_q^p$$

where $h(\gamma) = -\sum_{i,j} \gamma(i,j) \log(\gamma(i,j))$ is the entropy of $\gamma$ that allows to solve the transportation problem efficiently with the Sinkhorn-Knopp algorithm [18]. The second regularization term, where $\gamma(y_i = c, j)$ corresponds to the lines of the $j^{\text{th}}$ column of $\gamma$ where the class of the input is $c$, has been introduced in [17]. The goal of this term is to prevent input examples of different classes to move toward the same output examples by promoting group sparsity in the matrix $\gamma$ thanks to $\|\cdot\|_q^p$ corresponding to a $l_q$-norm to the power of $p$ used here with $q = 1$ and $p = \frac{1}{2}$.

### 3.3.2 Using a Class-based Representation Space

For this second approach, we propose to define virtual points as the unit vectors of a space of dimension $|\mathcal{Y}|$. Let $\mathbf{e}_j \in \mathbb{R}^{|\mathcal{Y}|}$ be such a unit vector ($1 \leq j \leq |\mathcal{Y}|$) -*i.e.* a vector where all the attributes are 0 except for one attribute $j$ which is set to 1- to which we associate a class label from $\mathcal{Y}$. Then, for any learning example $(\mathbf{x}_i, y_i)$, we define $f_\mathbf{v}(\mathbf{x}_i, y_i) = \mathbf{e}_{\#y_i}$ where $\#y_i = j$ if $\mathbf{e}_j$ is mapped with the class $y_i$. Thus, we have exactly $|\mathcal{Y}|$ virtual points, each one corresponding to a unit vector and a class label. We call this approach the class-based representation space method. If the number of classes is smaller than the number of dimensions used to represent the learning examples, then our method will perform dimensionality reduction for free. Furthermore, our approach will try to project all the examples of one class on the same axis while examples of other classes will tend to be projected on different axes. The underlying intuition behind the new space defined by $\mathbf{L}$ is to make each attribute discriminant for one class.

Table 1: Comparison of our approach with several baselines in the linear setting.

| Base | Baselines | | | Our approach | |
|---|---|---|---|---|---|
| | 1NN | LMNN | SCML | RVML-Lin-OT | RVML-Lin-Class |
| Amazon | 41.51 ± 3.24 | 65.50 ± 2.28 | 71.68 ± 1.86 | 71.62 ± 1.34 | **73.09 ± 2.49** |
| Breast | 95.49 ± 0.79 | 95.49 ± 0.89 | **96.50 ± 0.64*** | 95.24 ± 1.21 | 95.34 ± 0.95 |
| Caltech | 18.04 ± 2.20 | 49.68 ± 2.76 | 52.84 ± 1.61 | 52.51 ± 2.41 | **55.41 ± 2.55*** |
| DSLR | 29.61 ± 4.38 | **76.08 ± 4.79** | 65.10 ± 9.00 | 74.71 ± 5.27 | 75.29 ± 5.08 |
| Ionosphere | 86.23 ± 1.95 | 88.02 ± 3.02 | **90.38 ± 2.55*** | 87.36 ± 3.12 | 82.74 ± 2.81 |
| Isolet | 88.97 | **95.83** | 89.61 | 91.40 | 94.61 |
| Letters | 94.74 ± 0.27 | **96.43 ± 0.28*** | 96.13 ± 0.20 | 90.25 ± 0.60 | 95.51 ± 0.26 |
| Pima | 69.91 ± 1.69 | 70.04 ± 2.20 | 69.22 ± 2.60 | **70.48 ± 3.19** | 69.57 ± 2.85 |
| Scale | 78.68 ± 2.66 | 78.20 ± 1.91 | **93.39 ± 1.70*** | 90.05 ± 2.13 | 87.94 ± 1.99 |
| Splice | 71.17 | 82.02 | **85.43** | 84.64 | 78.44 |
| Svmguide1 | **95.12** | 95.03 | 87.38 | 94.83 | 85.25 |
| Wine | 96.18 ± 1.59 | 98.36 ± 1.03 | 96.91 ± 1.93 | **98.55 ± 1.67** | 98.18 ± 1.48 |
| Webcam | 42.90 ± 4.19 | 85.81 ± 3.75 | **90.43 ± 2.70** | 88.60 ± 3.63 | 88.60 ± 2.69 |
| mean | 69.89 | 82.81 | 83.46 | **83.86** | 83.07 |

# 4 Experimental results

In this section, we evaluate our approach on 13 different datasets coming from either the UCI [19] repository or used in recent works in metric learning [8, 20, 21]. For isolet, splice and svmguide1 we have access to a standard training/test partition, for the other datasets we use a 70% training/30% test partition, we perform the experiments on 10 different splits and we average the result. We normalize the examples with respect to the training set by subtracting for each attribute its mean and dividing by 3 times its standard deviation. We set our regularization parameter $\lambda$ with a 5-fold cross validation. After the metric learning step, we use a 1-nearest neighbor classifier to assess the performance of the metric and report the accuracy obtained.

We perform two series of experiments. First, we consider our linear formulation used with the two virtual points selection methods presented in this paper: RVML-Lin-OT based on Optimal Transport (Section 3.3.1) and RVML-Lin-Class using the class-based representation space method (Section 3.3.2). We compare them to a 1-nearest neighbor classifier without metric learning (1NN), and with two state of the art linear metric learning methods: LMNN [5] and SCML [20].
In a second series, we consider the kernelized versions of RVML, namely RVML-RBF-OT and RVML-RBF-Class, based respectively on Optimal Transport and class-based representation space methods, with a RBF kernel with the parameter $\sigma$ fixed as the mean of all pairwise training set Euclidean distances [16]. We compare them to non linear methods using a KPCA with a RBF kernel[2] as a pre-process: 1NN-KPCA a 1-nearest neighbor classifier without metric learning and LMNN-KPCA corresponding to LMNN in the KPCA-space. The number of dimensions is fixed as the one of the original space for high dimensional datasets (more than 100 attributes), to 3 times the original dimension when the dimension is smaller (between 5 and 100 attributes) and to 4 times the original dimension for the lowest dimensional datasets (less than 5 attributes). We also consider some local metric learning methods: GBLMNN [8] a non linear version of LMNN and SCMLLOCAL [20] the local version of SCML. For all these methods, we use the implementations available online letting them handle hyper-parameters tuning.

The results for linear methods are presented in Table 1 while Table 2 gives the results obtained with the non linear approaches. In each table, the best result on each line is highlighted with a bold font while the second best result is underlined. A star indicates either that the best baseline is significantly better than our best result or that our best result is significantly better than the best baseline according to classical significance tests (the p-value being fixed at $0.05$).

We can make the following remarks. In the linear setting, our approaches are very competitive to the state of the art and RVML-Lin-OT tends to be the best on average even though it must be noticed that SCML is very competitive on some datasets (the average difference is not significant). RVML-Lin-Class performs slightly less on average. Considering now the non linear methods, our approaches improve their performance and are significantly better than the others on average, RVML-RBF-Class has the best average behavior in this setting. These experiments show that our regressive formulation

Table 2: Comparison of our approach with several baselines in the non-linear case.

| | Baselines | | | | Our approach | |
|---|---|---|---|---|---|---|
| Base | 1NN-KPCA | LMNN-KPCA | GBLMNN | SCMLLOCAL | RVML-RBF-OT | RVML-RBF-Class |
| Amazon | $20.27 \pm 2.42$ | $53.16 \pm 3.73$ | $65.53 \pm 2.32$ | $69.14 \pm 1.74$ | $73.51 \pm 0.83$ | **76.22 ± 2.09*** |
| Breast | $92.43 \pm 2.19$ | $95.39 \pm 1.32$ | $95.58 \pm 0.87$ | **96.31 ± 0.66** | $95.73 \pm 0.97$ | $95.78 \pm 0.92$ |
| Caltech | $20.82 \pm 8.29$ | $29.88 \pm 10.89$ | $49.91 \pm 2.80$ | $50.56 \pm 1.62$ | $54.39 \pm 1.89$ | **57.98 ± 2.22*** |
| DSLR | $64.90 \pm 5.81$ | $73.92 \pm 7.57$ | $76.08 \pm 4.79$ | $62.55 \pm 6.94$ | $70.39 \pm 4.48$ | **76.67 ± 4.57** |
| Ionosphere | $75.57 \pm 2.79$ | $85.66 \pm 2.55$ | $87.36 \pm 3.02$ | $90.94 \pm 3.02$ | $90.66 \pm 3.10$ | **93.11 ± 3.30*** |
| Isolet | 68.70 | 96.28 | 96.02 | 91.40 | 95.96 | **96.73** |
| Letter | $95.39 \pm 0.27$ | **97.17* ± 0.18** | $96.51 \pm 0.25$ | $96.63 \pm 0.26$ | $91.26 \pm 0.50$ | $96.09 \pm 0.21$ |
| Pima | $69.57 \pm 2.64$ | $69.48 \pm 2.04$ | $69.52 \pm 2.27$ | $68.40 \pm 2.75$ | $69.35 \pm 2.95$ | **70.74 ± 2.36** |
| Scale | $78.36 \pm 0.88$ | $88.10 \pm 2.26$ | $77.88 \pm 2.43$ | $93.86 \pm 1.78$ | **95.19 ± 1.46*** | $94.07 \pm 2.02$ |
| Splice | 66.99 | **88.97** | 82.21 | 87.13 | 88.51 | 88.32 |
| Svmguide1 | **95.72** | 95.60 | 95.00 | 87.40 | 95.67 | 95.05 |
| Wine | $92.18 \pm 1.23$ | $95.82 \pm 2.98$ | $98.00 \pm 1.34$ | $96.55 \pm 2.00$ | **98.91 ± 1.53** | $98.00 \pm 1.81$ |
| Webcam | $73.55 \pm 4.57$ | $84.52 \pm 3.83$ | $85.81 \pm 3.75$ | $88.71 \pm 2.83$ | $88.71 \pm 4.28$ | **88.92 ± 2.91** |
| mean | 70.34 | 81.07 | 82.72 | 83.04 | 85.25 | **86.74** |

is very competitive and is even able to improve state of the art performances in a non linear setting and consequently that our virtual point selection methods automatically select correct instances.

Considering the virtual point selection, we can observe that the OT formulation performs better than the class-based representation space one in the linear case, while it is the opposite in the non-linear case. We think that this can be explained by the fact that the OT approach generates more virtual points in a potentially non linear way which brings more expressiveness for the linear case. On the other hand, in the non linear one, the relative small number of virtual points used by the class-based method seems to induce a better regularization. In Section 4 of the supplementary material, we provide additional experiments showing the interest of using explicit virtual points and the need of a careful association between examples and virtual points. We also provide some graphics showing 2D projections of the space learned by RVML-Lin-Class and RVML-RBF-Class on the Isolet dataset illustrating the capability of these approaches to learn discriminative attributes.

In terms of computational cost, our approach -implemented in closed form [22]- is competitive with classical methods but does not yield to significant improvements. Indeed, in practice, classical approaches only consider a small number of constraints *e.g.* $c$ times the number of examples, where $c$ is a small constant, in the case of SCML. Thus, the practical computational complexity of both our approach and classical methods is linearly dependant on the number of examples.

## 5 Conclusion

We present a new metric learning approach based on a regression and aiming at bringing closer the learning examples to some a priori defined virtual points. The number of constraints has the advantage to grow linearly with the size of the learning set in opposition to the quadratic grow of standard must-link cannot-link approaches. Moreover, our method can be solved in closed form and can be easily kernelized allowing us to deal with non linear problems. Additionally, we propose two methods to define the virtual points: One making use of recent advances in the field of optimal transport and one based on unit vectors of a class-based representation space allowing one to perform directly some dimensionality reduction. Theoretically, we show that our approach is consistent and we are able to link our empirical risk to the true risk of a classical metric learning formulation. Finally, we empirically show that our approach is competitive with the state of the art in the linear case and outperforms some classical approaches in the non-linear one.

We think that this work opens the door to design new metric learning formulations, in particular the definition of the virtual points can bring a way to control some particular properties of the metric (rank, locality, discriminative power, ...). As a consequence, this aspect opens new issues which are in part related to landmark selection problems but also to the ability to embed expressive semantic constraints to satisfy by means of the virtual points. Other perspectives include the development of a specific solver, of online versions, the use of low rank-inducing norms or the conception of new local metric learning methods. Another direction would be to study similarity learning extensions to perform linear classification such as in [21, 23].

## Footnotes

[1]When $\mathbf{M} = \mathbf{I}$, the identity matrix, it corresponds to the Euclidean distance.

[2]With the $\sigma$ parameter fixed as previously to the mean of all pairwise training set Euclidean distances.

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
