[Supplementary Material · supplementary.pdf]

# Regressive Virtual Metric Learning
# Supplementary Material

**Michaël Perrot, and Amaury Habrard**
Université de Lyon, Université Jean Monnet de Saint-Etienne,
Laboratoire Hubert Curien, CNRS, UMR5516, F-42000, Saint-Etienne, France.
{michael.perrot,amaury.habrard}@univ-st-etienne.fr

The goal of this supplementary is to present the proofs of the main theorems of the paper along the first three sections. Moreover, in Section 4 we provide additional experiments showing the interest of using explicit virtual points and the need of a careful association between examples and virtual points. We also provide some graphics showing 2D projections of the space learned by RVML-Lin-Class and RVML-RBF-Class on the Isolet dataset illustrating the capability of these approaches to learn discriminative attributes.

First of all, before presenting the proofs, we recall our setting for the sake of completeness. Given a probability distribution $\mathcal{D}$ defined over $\mathcal{X} \times \mathcal{Y}$ where $\mathcal{X} \subseteq \mathbb{R}^d$ and $\mathcal{Y}$ is a finite label set, let $S = \{(\mathbf{x}_i, y_i)\}_{i=1}^n$ be a set of examples drawn i.i.d. from $\mathcal{D}$. Let $f_{\mathbf{v}} : \mathcal{X} \times \mathcal{Y} \to \mathcal{V}$ where $\mathcal{V} \subseteq \mathbb{R}^{d'}$ be the function which associates each example to a virtual point such that $\mathbf{v} = f_{\mathbf{v}}(\mathbf{x}, y)$. We denote by $\mathcal{D}_{\mathbf{v}}$ the probability distribution defined on $\mathcal{X} \times \mathcal{V}$ obtained from the distribution $\mathcal{D}$ after applying $f_{\mathbf{v}}$, *i.e.* $p_{\mathcal{D}_{\mathbf{v}}}(\mathbf{x}, \mathbf{v}) = p_{\mathcal{D}}(\mathbf{x}, y | \mathbf{v} = f_{\mathbf{v}}(\mathbf{x}, y))$. Thus it is equivalent to obtain the set of examples $S_{\mathbf{v}} = \{(\mathbf{x}_i, \mathbf{v}_i)\}_{i=1}^n$ from $S$ after applying $f_{\mathbf{v}}$ and to draw $S_{\mathbf{v}}$ i.i.d. from $\mathcal{D}_{\mathbf{v}}$. Let $\| \cdot \|_{\mathcal{F}}$ be the Frobenius norm and $\| \cdot \|_2$ be the $l_2$ vector norm. We consider the following optimisation problem where we expanded the first Frobenius norm:

$$\mathbf{L} = \underset{\mathbf{L} \in \mathbb{R}^{d \times d'}}{\arg \min} f(\mathbf{L}) = \underset{\mathbf{L} \in \mathbb{R}^{d \times d'}}{\arg \min} \frac{1}{n} \sum_{(\mathbf{x},\mathbf{v}) \in S_{\mathbf{v}}} \|\mathbf{x}^T \mathbf{L} - \mathbf{v}^T\|_2^2 + \lambda \|\mathbf{L}\|_{\mathcal{F}}^2. \tag{1}$$

Furthermore, we define the loss (2), the empirical risk (3) and the true risk of our algorithm (4):

$$l(\mathbf{L}, (\mathbf{x}, \mathbf{v})) = \|\mathbf{x}^T \mathbf{L} - \mathbf{v}^T\|_2^2 \tag{2}$$

$$\hat{R}(\mathbf{L}) = \frac{1}{n} \sum_{(\mathbf{x},\mathbf{v}) \in S_{\mathbf{v}}} l(\mathbf{L}, (\mathbf{x}, \mathbf{v})) \tag{3}$$

$$R(\mathbf{L}) = \mathbb{E}_{(\mathbf{x},\mathbf{v}) \sim \mathcal{D}_{\mathbf{v}}} l(\mathbf{L}, (\mathbf{x}, \mathbf{v})) \tag{4}$$

## 1 Proof of Theorem 1

**Theorem 1.** *The optimal solution of Problem 1 can be found in closed form. Furthermore, we can derive two equivalent solutions:*

$$\mathbf{L} = \left(\mathbf{X}^T \mathbf{X} + \lambda n \mathbf{I}\right)^{-1} \mathbf{X}^T \mathbf{V} \tag{5}$$

$$\mathbf{L} = \mathbf{X}^T \left(\mathbf{X} \mathbf{X}^T + \lambda n \mathbf{I}\right)^{-1} \mathbf{V}. \tag{6}$$

*Proof.* Problem 1 is a classical regularized regression problem admitting a closed form solution [1]. We recall the derivation here for the sake of completeness. First we consider the derivative of

$f(\mathbf{L}, \mathbf{X}, \mathbf{V})$ with respect to $\mathbf{L}$:

$$\frac{\partial f(\mathbf{L}, \mathbf{X}, \mathbf{V})}{\partial \mathbf{L}} = 2\left(\frac{1}{n}\mathbf{X}^T\mathbf{X} + \lambda\mathbf{I}\right)\mathbf{L} - \frac{2}{n}\mathbf{X}^T\mathbf{V}.$$

Then we set this derivative to zero to obtain:

$$\mathbf{L} = \left(\mathbf{X}^T\mathbf{X} + \lambda n\mathbf{I}\right)^{-1}\mathbf{X}^T\mathbf{V}.$$

Finally Eq. 6 comes from using Taylor expansions as proposed in [1]. $\qquad\square$

## 2  Proof of Theorem 2

The interest of this theorem is to show that our algorithm is consistent, i.e. that with a sufficient number of examples the empirical risk tends to be close to the true risk. To prove this theorem we use the uniform stability framework presented in [2]. The idea is to show that changing one example in the training set does not change much the output of the algorithm. Thus, we start by upper bounding the Frobenius norm of $\mathbf{L}$ optimal solution of Problem 1 and the loss (2) considered. Afterwards, we show the $\sigma$-admissibility of the loss which allows us to prove the uniform stability of our algorithm which, in turns, allows us to apply Theorem 12 from [2].

In the following, we assume that $\|\mathbf{x}\|_2 \leq C_{\mathbf{x}}$ and $\|\mathbf{v}\|_2 \leq C_{\mathbf{v}}$. The next lemma upper bounds the Frobenius norm of $\mathbf{L}$ optimal solution of Problem 1:

**Lemma 1.** *Let $\mathbf{L}$ be an optimal solution of Problem 1, we have:*

$$\|\mathbf{L}\|_{\mathcal{F}} \leq \frac{C_{\mathbf{v}}}{\sqrt{\lambda}}.$$

*Proof.* Since $\mathbf{L}$ is an optimal solution of Problem 1, we have:

$$f(\mathbf{L}) \leq f(\mathbf{0})$$

$$\Leftrightarrow \quad \frac{1}{n}\sum_{(\mathbf{x},\mathbf{v})\in S_{\mathbf{v}}} l(\mathbf{L}, (\mathbf{x},\mathbf{v})) + \lambda\|\mathbf{L}\|_{\mathcal{F}}^2 \leq \frac{1}{n}\sum_{(\mathbf{x},\mathbf{v})\in S_{\mathbf{v}}} l(\mathbf{0}, (\mathbf{x},\mathbf{v})) + \lambda\|\mathbf{0}\|_{\mathcal{F}}^2$$

$$\Rightarrow \quad \lambda\|\mathbf{L}\|_{\mathcal{F}}^2 \leq \frac{1}{n}\sum_{(\mathbf{x},\mathbf{v})\in S_{\mathbf{v}}} \|\mathbf{v}\|_2^2 \qquad (7)$$

$$\Rightarrow \quad \lambda\|\mathbf{L}\|_{\mathcal{F}}^2 \leq C_{\mathbf{v}}^2$$

$$\Rightarrow \quad \|\mathbf{L}\|_{\mathcal{F}} \leq \frac{C_{\mathbf{v}}}{\sqrt{\lambda}}$$

Inequality 7 is obtained by noting that our loss is always positive. $\qquad\square$

We can now show that our loss is bounded.

**Lemma 2.** *The loss $l(\mathbf{L}, (\mathbf{x}, \mathbf{v}))$ is bounded by $M = C_{\mathbf{v}}^2\left(1 + \frac{C_{\mathbf{x}}}{\sqrt{\lambda}}\right)^2$.*

*Proof.*

$$l(\mathbf{L}, (\mathbf{x}, \mathbf{v})) = \|\mathbf{x}^T\mathbf{L} - \mathbf{v}^T\|_2^2$$

$$\leq \left(\|\mathbf{x}^T\|_2\|\mathbf{L}\|_{\mathcal{F}} + \|\mathbf{v}^T\|_2\right)^2 \qquad (8)$$

$$\leq \left(C_{\mathbf{x}}\frac{C_{\mathbf{v}}}{\sqrt{\lambda}} + C_{\mathbf{v}}\right)^2$$

$$\leq C_{\mathbf{v}}^2\left(1 + \frac{C_{\mathbf{x}}}{\sqrt{\lambda}}\right)^2.$$

Inequality 8 comes from the successive application of the triangle inequality and standard properties on norms. $\qquad\square$

We recall the definition of $\sigma$-admissibility from [2].

**Definition 1.** *A loss function $l$ is $\sigma$-admissible if it is convex with respect to its first argument and the following condition holds:*

$$\forall \mathbf{L}, \mathbf{L}' \in \mathbb{R}^{d \times d'}, \forall (\mathbf{x}, \mathbf{v}) \sim \mathcal{D}_{\mathbf{v}}, |l(\mathbf{L}, (\mathbf{x}, \mathbf{v})) - l(\mathbf{L}', (\mathbf{x}, \mathbf{v}))| \leq \sigma \|\mathbf{L} - \mathbf{L}'\|_{\mathcal{F}}$$

We show that our loss is $\sigma$-admissible in the following lemma.

**Lemma 3.** *The loss $l(\mathbf{L}, (\mathbf{x}, \mathbf{v}))$ is $\sigma$-admissible with $\sigma = 2C_{\mathbf{v}}C_{\mathbf{x}} \left(1 + \frac{C_{\mathbf{x}}}{\sqrt{\lambda}}\right)$.*

*Proof.*

$$\left| \|\mathbf{x}^T\mathbf{L}' - \mathbf{v}^T\|_2^2 - \|\mathbf{x}^T\mathbf{L}'' - \mathbf{v}^T\|_2^2 \right|$$
$$= \left| \|\mathbf{x}^T\mathbf{L}' - \mathbf{v}^T\|_2 - \|\mathbf{x}^T\mathbf{L}'' - \mathbf{v}^T\|_2 \right| \left| \|\mathbf{x}^T\mathbf{L}' - \mathbf{v}^T\|_2 + \|\mathbf{x}^T\mathbf{L}'' - \mathbf{v}^T\|_2 \right|$$
$$\leq \|\mathbf{x}^T\mathbf{L}' - \mathbf{v}^T - \mathbf{x}^T\mathbf{L}'' + \mathbf{v}^T\|_2 \left| \|\mathbf{x}^T\mathbf{L}' - \mathbf{v}^T\|_2 + \|\mathbf{x}^T\mathbf{L}'' - \mathbf{v}^T\|_2 \right| \qquad (9)$$
$$\leq \|\mathbf{L}' - \mathbf{L}''\|_{\mathcal{F}} 2C_{\mathbf{v}}C_{\mathbf{x}} \left(1 + \frac{C_{\mathbf{x}}}{\sqrt{\lambda}}\right). \qquad (10)$$

Inequality 9 is due to the reverse triangle inequality and inequality 10 follows from Lemma 2. ☐

We will now prove that our algorithm is uniformly stable but before we need the following lemma. In the following $\hat{R}(\mathbf{L})$ is the empirical risk over a set $S_{\mathbf{v}}$ of examples while we design by $\hat{R}^i(\mathbf{L})$ the empirical risk over a set $S_{\mathbf{v}}^i$ obtained from $S_{\mathbf{v}}$ by replacing its $i^{\text{th}}$ element. Similarly $f$ and $f^i$ denote the functions to optimize in Problem 1 using the sets of examples $S_{\mathbf{v}}$ and $S_{\mathbf{v}}^i$ respectively.

**Lemma 4.** *Let $f$ and $f^i$ be the functions to optimize, $\mathbf{L}$ and $\mathbf{L}^i$ their respective minimizers and $\lambda$ the regularization parameter used in our algorithm. Let $\Delta\mathbf{L} = \mathbf{L} - \mathbf{L}^i$, then, we have, for any $t \in [0, 1]$:*

$$\|\mathbf{L}\|_{\mathcal{F}}^2 - \|\mathbf{L} - t\Delta\mathbf{L}\|_{\mathcal{F}}^2 + \|\mathbf{L}^i\|_{\mathcal{F}}^2 - \|\mathbf{L}^i + t\Delta\mathbf{L}\|_{\mathcal{F}}^2 \leq \frac{4tC_{\mathbf{v}}C_{\mathbf{x}}}{\lambda n} \left(1 + \frac{C_{\mathbf{x}}}{\sqrt{\lambda}}\right) \|\Delta\mathbf{L}\|_{\mathcal{F}} \qquad (11)$$

*Proof.* This proof is similar to the proof in Lemma 20 in [2] which we recall here for the sake of completeness. First, note that $\hat{R}$ is a convex function, thus, for any $t \in [0, 1]$, we have:

$$\hat{R}^i(\mathbf{L} - t\Delta\mathbf{L}) - \hat{R}^i(\mathbf{L}) \leq t(\hat{R}^i(\mathbf{L}^i) - \hat{R}^i(\mathbf{L})) \qquad (12)$$
$$\hat{R}^i(\mathbf{L}^i + t\Delta\mathbf{L}) - \hat{R}^i(\mathbf{L}^i) \leq t(\hat{R}^i(\mathbf{L}) - \hat{R}^i(\mathbf{L}^i)) \qquad (13)$$

Summing inequalities (12) and (13) gives:

$$\hat{R}^i(\mathbf{L} - t\Delta\mathbf{L}) - \hat{R}^i(\mathbf{L}) + \hat{R}^i(\mathbf{L}^i + t\Delta\mathbf{L}) - \hat{R}^i(\mathbf{L}^i) \leq 0 \qquad (14)$$

$\mathbf{L}$ and $\mathbf{L}^i$ respectively minimize $f$ and $f^i$, we have:

$$f(\mathbf{L}) - f(\mathbf{L} - t\Delta\mathbf{L}) \leq 0 \qquad (15)$$
$$f^i(\mathbf{L}^i) - f^i(\mathbf{L}^i + t\Delta\mathbf{L}) \leq 0 \qquad (16)$$

Summing inequalities (14), (15) and (16) gives:

$$\hat{R}^i(\mathbf{L} - t\Delta\mathbf{L}) - \hat{R}^i(\mathbf{L}) + \hat{R}(\mathbf{L}) - \hat{R}(\mathbf{L} - t\Delta\mathbf{L})$$
$$+ \lambda\|\mathbf{L}\|_{\mathcal{F}}^2 - \lambda\|\mathbf{L} - t\Delta\mathbf{L}\|_{\mathcal{F}}^2 + \lambda\|\mathbf{L}^i\|_{\mathcal{F}}^2 - \lambda\|\mathbf{L}^i + t\Delta\mathbf{L}\|_{\mathcal{F}}^2 \leq 0. \qquad (17)$$

From (17), we can write:

$$\lambda\|\mathbf{L}\|_{\mathcal{F}}^2 - \lambda\|\mathbf{L} - t\Delta\mathbf{L}\|_{\mathcal{F}}^2 + \lambda\|\mathbf{L}^i\|_{\mathcal{F}}^2 - \lambda\|\mathbf{L}^i + t\Delta\mathbf{L}\|_{\mathcal{F}}^2 \leq B \qquad (18)$$

with

$$B = \hat{R}^i(\mathbf{L}) - \hat{R}^i(\mathbf{L} - t\Delta\mathbf{L}) + \hat{R}(\mathbf{L} - t\Delta\mathbf{L}) - \hat{R}(\mathbf{L}).$$

Using Lemma 3 we can bound B:

$$B \leq \left| \hat{R}^i(\mathbf{L}) - \hat{R}^i(\mathbf{L} - t\Delta\mathbf{L}) + \hat{R}(\mathbf{L} - t\Delta\mathbf{L}) - \hat{R}(\mathbf{L}) \right|$$

$$\leq \left| \frac{1}{n} \sum_{(\mathbf{x},\mathbf{v}) \in S_{\mathbf{v}}} l(\mathbf{L} - t\Delta\mathbf{L}, (\mathbf{x},\mathbf{v})) - \frac{1}{n} \sum_{(\mathbf{x},\mathbf{v})^i \in S_{\mathbf{v}}^i} l(\mathbf{L} - t\Delta\mathbf{L}, (\mathbf{x},\mathbf{v})^i) \right.$$

$$\left. + \frac{1}{n} \sum_{(\mathbf{x},\mathbf{v})^i \in S_{\mathbf{v}}^i} l(\mathbf{L}, (\mathbf{x},\mathbf{v})^i) - \frac{1}{n} \sum_{(\mathbf{x},\mathbf{v}) \in S_{\mathbf{v}}} l(\mathbf{L}, (\mathbf{x},\mathbf{v})) \right| \tag{19}$$

$$\leq \frac{1}{n} \left| l(\mathbf{L} - t\Delta\mathbf{L}, (\mathbf{x}_i, \mathbf{v}_i)) - l(\mathbf{L} - t\Delta\mathbf{L}, (\mathbf{x}_i, \mathbf{v}_i)^i) + l(\mathbf{L}, (\mathbf{x}_i, \mathbf{v}_i)^i) - l(\mathbf{L}, (\mathbf{x}_i, \mathbf{v}_i)) \right| \tag{20}$$

$$\leq \frac{1}{n} \left| l(\mathbf{L} - t\Delta\mathbf{L}, (\mathbf{x}_i, \mathbf{v}_i)) - l(\mathbf{L}, (\mathbf{x}_i, \mathbf{v}_i)) \right| + \frac{1}{n} \left| l(\mathbf{L}, (\mathbf{x}_i, \mathbf{v}_i)^i) - l(\mathbf{L} - t\Delta\mathbf{L}, (\mathbf{x}_i, \mathbf{v}_i)^i) \right|$$

$$\leq \frac{4tC_{\mathbf{v}}C_{\mathbf{x}}}{n} \left( 1 + \frac{C_{\mathbf{x}}}{\sqrt{\lambda}} \right) \|\Delta\mathbf{L}\|_{\mathcal{F}}. \tag{21}$$

Inequality 19 comes from the definition of the empirical risk and Inequality 20 is deduced by noting that the sums only differ by their $i^{\text{th}}$ element. Finally, we apply Lemma 3 twice to obtain Inequality 21. $\qquad\square$

We recall the definition of uniform stability [2] in the next definition.

**Definition 2.** *An algorithm A has uniform stability $\beta$ with respect to the loss function l if the following holds*

$$\forall S_{\mathbf{v}} \sim \mathcal{D}_{\mathbf{v}}^n, \forall i \in \{1, \ldots, n\}, \sup_{(\mathbf{x},\mathbf{v}) \sim \mathcal{D}_{\mathbf{v}}} \left| l(A_{S_{\mathbf{v}}}, (\mathbf{x},\mathbf{v})) - l(A_{S_{\mathbf{v}}^i}, (\mathbf{x},\mathbf{v})) \right| \leq \beta$$

*where $S_{\mathbf{v}}^i$ is a training set obtained from $S_{\mathbf{v}}$ when replacing its $i^{\text{th}}$ example with a new independent example and $A_{S_{\mathbf{v}}}$ and $A_{S_{\mathbf{v}}^i}$ stand for the optimal solution of algorithm A with respect to a given training set.*

**Lemma 5.** *Our algorithm has a uniform stability in $\beta = \frac{8C_{\mathbf{v}}^2 C_{\mathbf{x}}^2}{\lambda n} \left( 1 + \frac{C_{\mathbf{x}}}{\sqrt{\lambda}} \right)^2$.*

*Proof.* By setting $t = \frac{1}{2}$ in Lemma 4, one can obtain for the left hand side:

$$\|\mathbf{L}\|_{\mathcal{F}}^2 - \|\mathbf{L} - \frac{1}{2}\Delta\mathbf{L}\|_{\mathcal{F}}^2 + \|\mathbf{L}^i\|_{\mathcal{F}}^2 - \|\mathbf{L}^i + \frac{1}{2}\Delta\mathbf{L}\|_{\mathcal{F}}^2 = \frac{1}{2}\|\Delta\mathbf{L}\|_{\mathcal{F}}^2$$

and thus:

$$\frac{1}{2}\|\Delta\mathbf{L}\|_{\mathcal{F}}^2 \leq \frac{2C_{\mathbf{v}}C_{\mathbf{x}}}{\lambda n} \left( 1 + \frac{C_{\mathbf{x}}}{\sqrt{\lambda}} \right) \|\Delta\mathbf{L}\|_{\mathcal{F}}$$

$$\Rightarrow \qquad \|\Delta\mathbf{L}\|_{\mathcal{F}} \leq \frac{4C_{\mathbf{v}}C_{\mathbf{x}}}{\lambda n} \left( 1 + \frac{C_{\mathbf{x}}}{\sqrt{\lambda}} \right)$$

From Lemma 3 we have:

$$\left| l(\mathbf{L}, (\mathbf{x},\mathbf{v})) - l(\mathbf{L}^i, (\mathbf{x},\mathbf{v})) \right| \leq 2C_{\mathbf{v}}C_{\mathbf{x}} \left( 1 + \frac{C_{\mathbf{x}}}{\sqrt{\lambda}} \right) \|\Delta\mathbf{L}\|_{\mathcal{F}}$$

$$\leq \frac{8C_{\mathbf{v}}^2 C_{\mathbf{x}}^2}{\lambda n} \left( 1 + \frac{C_{\mathbf{x}}}{\sqrt{\lambda}} \right)^2$$

$$\square$$

We recall Theorem 12 from [2] for the sake of completeness:

**Theorem 12** ([2])**.** *Let A be an algorithm with uniform stability $\beta$ w.r.t. a loss function l such that $0 \leq l(A_{S_{\mathbf{v}}}, (\mathbf{x}, \mathbf{v})) \leq M$ for all $(\mathbf{x}, \mathbf{v}) \sim \mathcal{D}_{\mathbf{v}}$ and all sets $S_{\mathbf{v}}$. Then for any $n \geq 1$ the following bound holds with probability at least $1 - \delta$ over the random draw of the sample $S_{\mathbf{v}}$,*

$$R(A_{S_{\mathbf{v}}}) \leq \hat{R}(A_{S_{\mathbf{v}}}) + \beta + (2n\beta + M)\sqrt{\frac{\ln \frac{1}{\delta}}{2n}}.$$

We have shown that our algorithm is uniformly stable and that our loss is bounded, hence we can apply this theorem to get Theorem 2.

**Theorem 2.** *Let $\|\mathbf{v}\|_2 \leq C_{\mathbf{v}}$ for any $\mathbf{v} \in \mathcal{V}$ and $\|\mathbf{x}\|_2 \leq C_{\mathbf{x}}$ for any $\mathbf{x} \in \mathcal{X}$. With probability $1 - \delta$, for any matrix $\mathbf{L}$ optimal solution of Problem 1, we have:*

$$R(\mathbf{L}) \leq \hat{R}(\mathbf{L}) + \frac{8C_{\mathbf{v}}^2 C_{\mathbf{x}}^2}{\lambda n}\left(1 + \frac{C_{\mathbf{x}}}{\sqrt{\lambda}}\right)^2 + \left(\left(\frac{16C_{\mathbf{x}}^2}{\lambda} + 1\right)C_{\mathbf{v}}^2\left(1 + \frac{C_{\mathbf{x}}}{\sqrt{\lambda}}\right)^2\right)\sqrt{\frac{\ln \frac{1}{\delta}}{2n}}.$$

*Proof.* This theorem is a direct application of Theorem 12 from [2] using the bound on the loss presented in Lemma 2 and the uniform stability of our algorithm proven in Lemma 5. $\square$

**Kernelized case** Recall that in the linear case we assumed that $\|\mathbf{x}\|_2 \leq C_{\mathbf{x}}$. In the kernelized case, we only have to bound $\|\phi(\mathbf{x})\|_2$ where $\phi$ is the projection function associated to the used kernel. A common assumption [3] when studying kernels is that $\exists \kappa$ such that $0 < \kappa < \infty$ and $K(\mathbf{x}, \mathbf{x}) \leq \kappa^2$. Hence, we have $\|\phi(\mathbf{x})\|_2^2 \leq \kappa^2$. Thus setting $C_{\mathbf{x}} = \kappa$ allows us to use the same proof than in the linear case leading us to the same generalization bound (the only difference being the value of $C_{\mathbf{x}}$).

## 3 Proof of Theorem 3

For the sake of readability we recall the loss for the classical metric learning approach [4] considered here:

$$l(\mathbf{L}, (\mathbf{x}_i, y_i), (\mathbf{x}_j, y_j)) = \left[y_{ij}(d^2(\mathbf{L}^T\mathbf{x}_i, \mathbf{L}^T\mathbf{x}_j) - \gamma_{y_{ij}})\right]_+ \tag{22}$$

and the theorem:

**Theorem 3.** *Let $\mathcal{D}$ be a distribution over $\mathcal{X} \times \mathcal{Y}$. Let $\mathcal{V} \subset \mathbb{R}^{d'}$ be a finite set of virtual points and $f_{\mathbf{v}}$ is defined as $f_{\mathbf{v}}(\mathbf{x}_i, y_i) = \mathbf{v}_i$, $\mathbf{v}_i \in \mathcal{V}$. Let $\|\mathbf{v}\|_2 \leq C_{\mathbf{v}}$ for any $\mathbf{v} \in \mathcal{V}$ and $\|\mathbf{x}\|_2 \leq C_{\mathbf{x}}$ for any $\mathbf{x} \in \mathcal{X}$. Let $\gamma_1 = 2 \max_{\mathbf{x}_k, \mathbf{x}_l, y_{kl}=1} d^2(\mathbf{v}_k, \mathbf{v}_l)$ and $\gamma_{-1} = \frac{1}{2} \min_{\mathbf{x}_k, \mathbf{x}_l, y_{kl}=-1} d^2(\mathbf{v}_k, \mathbf{v}_l)$, we have:*

$$\mathbb{E}_{(\mathbf{x}_i, y_i) \sim \mathcal{D}, (\mathbf{x}_j, y_j) \sim \mathcal{D}} \left[y_{ij}(d^2(\mathbf{L}^T\mathbf{x}_i, \mathbf{L}^T\mathbf{x}_j) - \gamma_{y_{ij}})\right]_+$$

$$\leq 8\left(\hat{R}(\mathbf{L}) + \frac{8C_{\mathbf{v}}^2 C_{\mathbf{x}}^2}{\lambda n}\left(1 + \frac{C_{\mathbf{x}}}{\sqrt{\lambda}}\right)^2 + \left(\left(\frac{16C_{\mathbf{x}}^2}{\lambda} + 1\right)C_{\mathbf{v}}^2\left(1 + \frac{C_{\mathbf{x}}}{\sqrt{\lambda}}\right)^2\right)\sqrt{\frac{\ln \frac{1}{\delta}}{2n}}\right).$$

*Proof.* First of all, let us consider two examples $\mathbf{x}_i$ and $\mathbf{x}_j$ and their associated virtual points $\mathbf{v}_i$ and $\mathbf{v}_j$.

Using the fact that distances respect the triangle inequality, one can obtain:

$$d(\mathbf{L}^T\mathbf{x}_i, \mathbf{L}^T\mathbf{x}_j) \leq d(\mathbf{L}^T\mathbf{x}_i, \mathbf{v}_i) + d(\mathbf{v}_i, \mathbf{v}_j) + d(\mathbf{v}_j, \mathbf{L}^T\mathbf{x}_j).$$

Then squaring both sides of the inequality gives:

$$d^2(\mathbf{L}^T\mathbf{x}_i, \mathbf{L}^T\mathbf{x}_j) \leq d^2(\mathbf{L}^T\mathbf{x}_i, \mathbf{v}_i) + d^2(\mathbf{v}_i, \mathbf{v}_j) + d^2(\mathbf{v}_j, \mathbf{L}^T\mathbf{x}_j)$$
$$+ 2(d(\mathbf{L}^T\mathbf{x}_i, \mathbf{v}_i) + d(\mathbf{v}_j, \mathbf{L}^T\mathbf{x}_j))d(\mathbf{v}_i, \mathbf{v}_j) + 2d(\mathbf{L}^T\mathbf{x}_i, \mathbf{v}_i)d(\mathbf{v}_j, \mathbf{L}^T\mathbf{x}_j).$$

Finally, using Legendre identity[1] twice, we obtain:

$$d^2(\mathbf{L}^T\mathbf{x}_i, \mathbf{L}^T\mathbf{x}_j) \leq 4d^2(\mathbf{L}^T\mathbf{x}_i, \mathbf{v}_i) + 2d^2(\mathbf{v}_i, \mathbf{v}_j) + 4d^2(\mathbf{v}_j, \mathbf{L}^T\mathbf{x}_j).$$

Similarly, switching the role of $d(\mathbf{L}^T\mathbf{x}_i, \mathbf{L}^T\mathbf{x}_j)$ and $d(\mathbf{v}_i, \mathbf{v}_j)$ we have:

$$d^2(\mathbf{v}_i, \mathbf{v}_j) \leq 4d^2(\mathbf{L}^T\mathbf{x}_i, \mathbf{v}_i) + 2d^2(\mathbf{L}^T\mathbf{x}_i, \mathbf{L}^T\mathbf{x}_j) + 4d^2(\mathbf{v}_j, \mathbf{L}^T\mathbf{x}_j)$$

$$\Leftrightarrow \quad -d^2(\mathbf{L}^T\mathbf{x}_i, \mathbf{L}^T\mathbf{x}_j) \leq 2d^2(\mathbf{L}^T\mathbf{x}_i, \mathbf{v}_i) + 2d^2(\mathbf{v}_j, \mathbf{L}^T\mathbf{x}_j) - \frac{1}{2}d^2(\mathbf{v}_i, \mathbf{v}_j)$$

$$\Leftrightarrow \quad -d^2(\mathbf{L}^T\mathbf{x}_i, \mathbf{L}^T\mathbf{x}_j) \leq 4d^2(\mathbf{L}^T\mathbf{x}_i, \mathbf{v}_i) + 4d^2(\mathbf{v}_j, \mathbf{L}^T\mathbf{x}_j) - \frac{1}{2}d^2(\mathbf{v}_i, \mathbf{v}_j)$$

Now, let us consider two examples of the same class, *i.e.* $y_{ij} = 1$, we have:

$$\left[y_{ij}(d^2(\mathbf{L}^T\mathbf{x}_i, \mathbf{L}^T\mathbf{x}_j) - \gamma_{y_{ij}})\right]_+ = \left[d^2(\mathbf{L}^T\mathbf{x}_i, \mathbf{L}^T\mathbf{x}_j) - \gamma_1\right]_+$$
$$\leq \left[4d^2(\mathbf{L}^T\mathbf{x}_i, \mathbf{v}_i) + 4d^2(\mathbf{v}_j, \mathbf{L}^T\mathbf{x}_j) + 2d^2(\mathbf{v}_i, \mathbf{v}_j) - \gamma_1\right]_+$$
$$\leq 4d^2(\mathbf{L}^T\mathbf{x}_i, \mathbf{v}_i) + 4d^2(\mathbf{v}_j, \mathbf{L}^T\mathbf{x}_j). \tag{23}$$

Inequality 23 comes from the fact that $\gamma_1 \geq 2d^2(\mathbf{v}_i, \mathbf{v}_j)$ and by noting that a distance is always positive.

Similarly, we consider two examples of different classes, *i.e.* $y_{ij} = -1$, and we obtain:

$$\left[y_{ij}(d^2(\mathbf{L}^T\mathbf{x}_i, \mathbf{L}^T\mathbf{x}_j) - \gamma_{y_{ij}})\right]_+ = \left[-d^2(\mathbf{L}^T\mathbf{x}_i, \mathbf{L}^T\mathbf{x}_j) + \gamma_{-1}\right]_+$$
$$\leq \left[4d^2(\mathbf{L}^T\mathbf{x}_i, \mathbf{v}_i) + 4d^2(\mathbf{v}_j, \mathbf{L}^T\mathbf{x}_j) - \frac{1}{2}d^2(\mathbf{v}_i, \mathbf{v}_j) + \gamma_{-1}\right]_+$$
$$\leq 4d^2(\mathbf{L}^T\mathbf{x}_i, \mathbf{v}_i) + 4d^2(\mathbf{v}_j, \mathbf{L}^T\mathbf{x}_j). \tag{24}$$

Inequality 24 comes from the fact that $\gamma_{-1} \leq \frac{1}{2}d^2(\mathbf{v}_i, \mathbf{v}_j)$ and by noting that a distance is always positive.

Taking the expectation on both sides of Inequality 24 gives:

$$\mathbb{E}_{(\mathbf{x}_i, y_i) \sim \mathcal{D}, (\mathbf{x}_j, y_j) \sim \mathcal{D}} \left[y_{ij}(d^2(\mathbf{L}^T\mathbf{x}_i, \mathbf{L}^T\mathbf{x}_j) - \gamma_{y_{ij}})\right]_+ \tag{25}$$
$$\leq \mathbb{E}_{(\mathbf{x}_i, y_i) \sim \mathcal{D}, (\mathbf{x}_j, y_j) \sim \mathcal{D}} 4d^2(\mathbf{L}^T\mathbf{x}_i, \mathbf{v}_i) + 4d^2(\mathbf{v}_j, \mathbf{L}^T\mathbf{x}_j)$$
$$\leq \mathbb{E}_{(\mathbf{x}_i, y_i) \sim \mathcal{D}, (\mathbf{x}_j, y_j) \sim \mathcal{D}} 4d^2(\mathbf{L}^T\mathbf{x}_i, \mathbf{v}_i) + \mathbb{E}_{(\mathbf{x}_i, y_i) \sim \mathcal{D}, (\mathbf{x}_j, y_j) \sim \mathcal{D}} 4d^2(\mathbf{v}_j, \mathbf{L}^T\mathbf{x}_j)$$
$$\leq 8\mathbb{E}_{(\mathbf{x}, y) \sim \mathcal{D}} d^2(\mathbf{L}^T\mathbf{x}, \mathbf{v})$$
$$\leq 8R(\mathbf{L}).$$

Applying Theorem 2 to the last inequality gives the theorem. □

# 4 Extended Experiments

In this section, we propose several experiments showing the interest of using explicit virtual points and the need of a careful association between examples and virtual points. We also provide some graphics showing 2D projections of the space learned by RVML-Lin-Class and RVML-RBF-Class on the isolet dataset illustrating the capability of these approaches to learn discriminative attributes.

## 4.1 Interest of Explicit Virtual Points

In [5] the authors propose to collapse similar examples on a single point, an implicit virtual point, while pushing far away dissimilar examples. This behavior can, in fact, be achieved by any margin based metric learning approach by setting the margin between similar examples to 0 and the margin between dissimilar examples to a high value. Thus to illustrate the interest of using explicit virtual points, we propose to compare our approach to ITML when considering the aforementioned margins (ITML-Collapse). For the sake of completeness we also consider ITML with tuned margins (ITML). The results are presented in Table 1 and show that, on average, ITML-Collapse is less accurate than RVML-Lin-Class hinting that considering explicit virtual points is better than considering implicit ones.

Table 1: Comparison between a method with explicit virtual points (RVML-Lin-Class) and a method with implicit virtual points (ITML-Collapse).

| Base | RVML-Lin-Class | ITML-Collapse | ITML |
|---|---|---|---|
| Amazon | **73.09 $\pm$ 2.49** | 57.97 $\pm$ 3.36 | 65.91 $\pm$ 2.64 |
| Breast | 95.34 $\pm$ 0.95 | 94.56 $\pm$ 1.41 | **95.49 $\pm$ 1.15** |
| Caltech | **55.41 $\pm$ 2.55** | 37.34 $\pm$ 2.01 | 47.31 $\pm$ 2.75 |
| DSLR | 75.29 $\pm$ 5.08 | **77.25 $\pm$ 4.15** | 77.25 $\pm$ 4.91 |
| Ionosphere | 82.74 $\pm$ 2.81 | 85.75 $\pm$ 6.23 | **88.11 $\pm$ 1.68** |
| Isolet | **94.61** | 74.53 | 92.88 |
| Letters | 95.51 $\pm$ 0.26 | **95.67 $\pm$ 0.30** | 95.00 $\pm$ 0.64 |
| Pima | 69.57 $\pm$ 2.85 | **71.08 $\pm$ 2.13** | 70.26 $\pm$ 1.38 |
| Scale | **87.94 $\pm$ 1.99** | 87.51 $\pm$ 4.39 | 87.67 $\pm$ 2.71 |
| Splice | **78.44** | 66.80 | 71.49 |
| Svmguide1 | 85.25 | 94.62 | **95.00** |
| Wine | **98.18 $\pm$ 1.48** | 85.91 $\pm$ 3.74 | 96.91 $\pm$ 1.93 |
| Webcam | 88.60 $\pm$ 2.69 | **97.64 $\pm$ 2.43** | 86.56 $\pm$ 2.88 |
| mean | **83.07** | 78.97 | 82.30 |

Table 2: Comparison of our OT based formulation to a random selection approach when learning a linear metric.

| | OT based approach | Random | | |
|---|---|---|---|---|
| Base | RVML-Lin-OT | 1 VP per class | 2 VP per class | 5 VP per class |
| Amazon | 71.62 $\pm$ 1.34 | **74.23 $\pm$ 2.15** | 72.92 $\pm$ 2.31 | 70.31 $\pm$ 2.82 |
| Breast | 95.24 $\pm$ 1.21 | **95.34 $\pm$ 0.95** | 95.29 $\pm$ 1.32 | 94.90 $\pm$ 1.92 |
| Caltech | 52.51 $\pm$ 2.41 | **55.09 $\pm$ 2.38** | 53.63 $\pm$ 2.12 | 49.59 $\pm$ 1.69 |
| DSLR | **74.71 $\pm$ 5.27** | 70.59 $\pm$ 6.06 | 63.53 $\pm$ 5.08 | 52.16 $\pm$ 8.68 |
| Ionosphere | 87.36 $\pm$ 3.12 | 82.74 $\pm$ 2.81 | 88.40 $\pm$ 4.05 | **90.28 $\pm$ 3.33** |
| Isolet | 91.40 | 92.75 | **94.16** | 92.43 |
| Letters | 90.25 $\pm$ 0.60 | 89.90 $\pm$ 1.02 | 90.54 $\pm$ 1.24 | **91.13 $\pm$ 0.74** |
| Pima | **70.48 $\pm$ 3.19** | 69.57 $\pm$ 2.85 | 69.35 $\pm$ 2.44 | 69.26 $\pm$ 2.60 |
| Scale | **90.05 $\pm$ 2.13** | 88.10 $\pm$ 2.57 | 89.47 $\pm$ 2.99 | 89.21 $\pm$ 2.68 |
| Splice | **84.64** | 78.44 | 78.94 | 80.87 |
| Svmguide1 | **94.83** | 85.25 | 86.90 | 94.70 |
| Wine | **98.55 $\pm$ 1.67** | 98.55 $\pm$ 1.43 | 97.64 $\pm$ 2.43 | 98.00 $\pm$ 1.34 |
| Webcam | 88.60 $\pm$ 3.63 | **88.92 $\pm$ 3.21** | 86.24 $\pm$ 2.95 | 81.18 $\pm$ 3.56 |
| mean | **83.86** | 82.27 | 82.08 | 81.08 |

## 4.2 Association of Examples and Virtual Points

To further assess the interest of using our OT based formulation to select virtual points and associate them to examples, we propose to compare it with a random based approach (Random). In this latter setting, we randomly select a subset of examples for each class to act as virtual points and we randomly associate each example of this class to these virtual points. The results in the linear case are presented in Table 2 while the results in the non-linear case are presented in Table 3. Overall, randomly selecting the virtual points is less accurate than using the OT based formulation. This is especially true in the linear case where the metric is less expressive than in the kernelized case and thus requires more meaningful virtual points. Hence, selecting virtual points and correctly associating them to the examples is key to obtain a good performance.

Table 3: Comparison of our OT based formulation to a random selection approach when learning a non linear metric.

| | OT based approach | Random | | |
|---|---|---|---|---|
| Base | RVML-RBF-OT | 1 VP per class | 2 VP per class | 5 VP per class |
| Amazon | $73.51 \pm 0.83$ | **$75.74 \pm 2.35$** | $72.68 \pm 2.02$ | $70.07 \pm 2.86$ |
| Breast | $95.73 \pm 0.97$ | $95.73 \pm 1.07$ | **$95.83 \pm 0.80$** | $95.58 \pm 1.38$ |
| Caltech | $54.39 \pm 1.89$ | **$58.33 \pm 2.05$** | $53.98 \pm 3.18$ | $50.35 \pm 1.89$ |
| DSLR | **$70.39 \pm 4.48$** | $65.29 \pm 7.51$ | $58.24 \pm 7.79$ | $48.82 \pm 8.03$ |
| Ionosphere | **$90.66 \pm 3.10$** | $90.57 \pm 3.05$ | $89.25 \pm 3.73$ | $90.38 \pm 3.26$ |
| Isolet | $95.96$ | **$96.99$** | $96.54$ | $95.25$ |
| Letters | $91.26 \pm 0.50$ | $91.77 \pm 0.43$ | $91.87 \pm 0.52$ | **$92.04 \pm 0.62$** |
| Pima | $69.35 \pm 2.95$ | $70.82 \pm 4.60$ | **$71.26 \pm 2.84$** | $70.00 \pm 2.56$ |
| Scale | **$95.19 \pm 1.46$** | $93.39 \pm 2.19$ | $91.96 \pm 1.69$ | $91.32 \pm 1.95$ |
| Splice | **$88.51$** | $88.37$ | $88.46$ | $87.22$ |
| Svmguide1 | **$95.67$** | $95.03$ | $95.55$ | $95.88$ |
| Wine | **$98.91 \pm 1.53$** | $97.82 \pm 1.88$ | $97.27 \pm 1.96$ | $97.82 \pm 1.67$ |
| Webcam | **$88.71 \pm 4.28$** | $87.31 \pm 2.99$ | $83.01 \pm 3.28$ | $76.67 \pm 4.78$ |
| mean | **$85.25$** | $85.17$ | $83.53$ | $81.65$ |

### 4.3  Illustration of the Behavior of Our Approach on One Dataset

To illustrate the capability of RVML-Lin-Class and RVML-RBF-Class to learn discriminative attributes we propose to select two dimensions out of the 26 of the space learned by these approaches on the isolet dataset. We selected 3 pairs of axis and the images obtained are presented in Fig. 1. On the same line, we plot two images corresponding to the same axis pair: on the left column for RVML-Lin-Class and on the right column for RVML-RBF-Class. Note that for each axis, there is only one class for which the value of the attribute tends to be 1, for all the other classes this feature tends to be 0. Furthermore, we can note that the kernelized version of our metric outputs a more discriminative space: the examples are brought closer to their corresponding virtual point than in the linear version.

## Footnotes

[1]Legendre identity is $(a + b)^2 - (a - b)^2 = 4ab$ from which we deduce $a^2 + b^2 \geq 2ab$.

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

Figure 1: In the learned space from the isolet dataset, we randomly select 2 attributes three times and plot the 2D projection on each pair. The first line corresponds to features 1 and 20, the second line to features 7 and 14 and the third line to features 2 and 23. The left column corresponds to the space learned by RVML-Lin-Class (linear) and the right column to the one learned by RVML-RBF-Class (non linear). (Best viewed in color)