[Reviews · NeurIPS 2015]

Submitted by Assigned_Reviewer_1

The formulation is simple but seems effective. The way to learn the dirtual points is critical, but motivation of the use of optimal transport is not clearly stated.
Summary: This paper proposes a way to learn Mahalanobis distance metric by minimizing the transformed distance of samples to certain predefined virtual points.

Submitted by Assigned_Reviewer_2

The paper proposes to learn a distance metric such that the examples in a dataset are moved closer to their corresponding virtual points. Virtual points are either landmarks selected from the training set or some representation of the label space. Closed form solutions for the distance matrix is obtained for both linear and kernalized cases along with several theoretical guarantees on consistency and relations to classical metric learning. Experiments are performed on several UCI datasets and relevant comparisons to previous state-of-art shows promise for the proposed approach.

+ The main idea of learning a metric that moves points closer to common virtual points seems novel and experiments do justify the idea to a large extent. - The paper has several typos and grammatical errors and could improve on presentation.
Summary: The main contribution of the paper is a new metric learning approach that moves examples belonging to different classes to different virtual points along with a kernalized version which is a novel idea. Experiments on several datasets show improvements over existing metric learning approaches using both the linear and kernalized methods.

Submitted by Assigned_Reviewer_3

Summary: This paper proposed a novel approach for efficient metric learning. The objective of classic metric learning is to solve the positive semi-definite matrix $M$ such that the distance between examples from the same class get closer while the distance between examples from different classes get further in the distance define by M. Unlike these methods, this paper proposed a idea of virtual point-based regression formulation. The points from the same class are pulled toward the virtual points, thus their distance to each other become closer and their distance to points in other classes become further. The paper describes a closed form solution to compute the regression matrix and two different ways to discover the virtual points. The method is evaluated in 13 different metric learning datasets and compared with several standard baselines.

Pros: - The experiment result is complete and supports the proposed method. It is evaluated in 13 different datasets, linear and RBF kernel, and compared with several baselines. - When learning the metric, the number of constraints only grows with number of virtual points, which is linear to the training size. For existing metric learning, the number of constraints is quadratic to the training size. - The paper is well written and easy to follow. The introduction and related work clearly explains the novelty and key contribution of the proposed method clearly. Supplementary material provides detail explanation of theorem.

Cons: - If I understand it correctly, the selection process described in in Algorithm 1 is quadratic to the size of S because of two argmax over x? - The performance of the proposed method doesn't work well with linear kernel.

Suggestion: - Since the proposed method requires less constraints, the learning process should be more efficient than other methods. It would be helpful if we could also see some results for the training time comparison against other methods.

- Compare to LMNN, could the proposed method provide additional advantage in adaptive(online) metric learning? - To further demonstrate the benefit of OT on learning set, the author could consider adding the baseline by randomly selecting an example from each class as virtual point.
Summary: This paper proposed a novel approach for efficient metric learning. The key idea is to push the data points from the same class to one(or more) virtual point(s). The proposed formulation consider less constraints than common metric learning methods. The method is thoroughly tested in 13 different datasets and shows improvement over baselines.

Submitted by Assigned_Reviewer_4

The authors propose to do the supervised metric learning by introducing virtual points and then move the examples close to the virtual point associated with each example. The cannot link constraints are not explicitly enforced but rather implicitly in the formulation. As a result, the number of constraints are linear in number of examples rather than quadratic. The approach has a closed form solution and can be easily kernelized. The authors also propose two ways of finding the virtual points.

The quality of the work is good. All aspects of the method are clearly explained and are elegant. The experiments section show good improvement in the non-linear case. My reason for giving relatively lower rating is because I think one important comparison with [6] is missing. The main difference between [6] and this work is possibility of use of multiple virtual points per class which makes it important to show the comparison. Also paper claims the linearity wrt. number of examples as advantage. But no timing results are given.

The paper is mostly clear to read.

The work has slight novelty in terms of using multiple virtual points than single one per class, although this aspect is not evaluated properly in the experiments.
Summary: The idea of introducing multiple virtual points is new and the authors reduce the complexity from quadratic to linear and to a simple regression one with closed form solution and direct and amenable to Kernelizaton. Results show improved performance in kernel space, however comparison with single virtual point method [6] is missing.

Author Feedback
Author rebuttal: We thank all the reviewers for the quality of their feedback. Please find our answers to specific concerns below.

-R1: Thanks we will carefully proofread the paper with a native english speaker.

-R2,R3,R4: About optimal transport (OT) and one virtual point per class
The motivation for using OT is maybe not sufficiently clear, thank you for having stressed this, we will remedy to this drawback.
As suggested by R2, selecting randomly one example per class as a virtual point has the following behavior. In the linear case, an average accuracy of 82.26 is achieved which is less accurate than other baselines. It shows that randomly selecting a virtual point is not sufficient for linear models, while using OT allows us to obtain more meaningful virtual points. In the nonlinear case, the accuracy improves to 85.17 and the difference with the OT approach is smaller. Indeed, using a kernel improves the expressiveness of the metric, thus reducing the importance of selecting meaningful virtual points. We will add these results to the paper and comment them adequately.
Hence, using OT allows us (i) to select meaningful virtual points in an automatic way, (ii) to associate each example with the best possible virtual point, (iii) to bring more expressiveness by using multiple virtual points notably for linear settings. Furthermore, we considered an OT formulation that makes use of some labels, however one could imagine using other kind of supervision (or no supervision with the original OT) to map each example to a virtual point. We think that it offers interesting perspectives.

-R2,R3: About running times
While most metric learning formulations consider a quadratic number of constraints, it is hard to deal with all them in practice. A common strategy is to select only a subset of constraints. For example, SCML implementation considers c*n triplets where c is the product between the number of neighbors and the number of impostors considered. Thus they restrain to a linear number of constraints. Similarly, LMNN restricts to k neighbors and to impostors lying in a ball around the targeted instance. As a consequence, when n is large, our approaches are not significantly faster but are still comparable to SCML or LMNN. When n is smaller (~1000) our approaches tend to be faster by a factor between 2 and 10.
Note that as R2 noticed, Alg.1 is quadratic in the number of examples, more precisely O(|S|*|S'|) but usually |S'| is far lower than |S|. Hence, even if it slows down our algorithm, it remains competitive.
Finally, our framework can be seen as a well founded method for dealing with a linear number of constraints with an expressiveness comparable to methods with a quadratic number of constraints (see Th3), or in other words it justifies that a linear number of constraints can be sufficient if they are chosen adequately wrt to virtual points.
We will clarify these aspects in the paper.

-R2: About an online extension
It is indeed promising. The main advantage would be that we do not require examples to arrive by pairs anymore. Indeed, given a new example and the mapping function f_v() we believe that it is possible to update the metric by adapting known online regression tools.

-R2,R7: About the results (linear case)
With the linear kernel our approach is competitive with state of the art approaches. Indeed, if we do not always obtain the best result on each dataset, we are better on average (RVML-Lin-OT). It illustrates a kind of stability while being accurate. Additionally, the improvement is significant with kernelized versions.

-R3: About comparison to [6]
We agree that this comparison could be interesting, we have actually focused on methods with an implementation available online (we did not find any for [6]). However, we think that it is not a critical issue. Indeed, in [6], the authors do not define an explicit notion of virtual point; this is actually a consequence of the formulation that tries to achieve a null distance between examples of the same class, which implies to collapse them in a single point while requiring a quadratic number of constraints.
In fact, this idea of collapsing all similar examples in a single point can be achieved by any margin based metric learning method by fixing the margin to 0 between similar examples. We have performed some tests in this setting; it does not change the line of the results, even though tuning the margin adequately tends to produce better results. As an illustration, consider ITML that was compared to [6] in the original paper [1] (this paper reports better results for ITML on average). Applying ITML on the datasets of our paper it achieves an average accuracy of 82.30. Then, fixing the margin to the lowest possible value in ITML further lower the results to less than 81 hinting that using explicit virtual points is better. We propose to include these results in the paper.
Note that we will also publish our code online.